# Effects of Repetitive Transcranial Magnetic Stimulation on Gait and Postural Control Ability of Patients with Executive Dysfunction after Stroke

**DOI:** 10.3390/brainsci12091185

**Published:** 2022-09-02

**Authors:** Huixian Yu, Sihao Liu, Pei Dai, Zhaoxia Wang, Changbin Liu, Hao Zhang

**Affiliations:** 1Department of Rehabilitation Medicine, Beijing Tiantan Hospital, Capital Medical University, Beijing 100060, China; 2School of Rehabilitation, China Rehabilitation Research Center, Capital Medical University, Beijing 100068, China

**Keywords:** stroke, executive function, postural control, gait, repetitive transcranial stimulation

## Abstract

Objective: To assess the effects of repetitive transcranial magnetic stimulation (rTMS) on the gait and postural control ability of patients with executive dysfunction (ED) after stroke. Methods: A total of 18 patients with ED after stroke were randomly assigned into two groups, including an experimental group and a sham group. Patients in both groups received routine rehabilitation therapy, and patients in the experimental group underwent rTMS on the left dorsolateral prefrontal cortex (DLPFC) for 2 weeks (5 HZ, 80%MT, 1200 pulses). In the sham group, patients experienced sham stimulation treatment, in which the coil was placed vertically with the head. Before and after treatment, patients in both groups were subjected to Montreal cognitive assessment (MoCA) scoring, Fugl–Meyer assessment of lower extremity (L-FMA), Stroop color-word test (SCWT), gait analysis, foot plantar pressure test, 10-m walking test (10MWT), Berg balance scale (BBS), and timed up and go test (TUGT). In the SCWT, it was attempted to record the time of each card (SCWT-T), the correct number (SCWT-C), Stroop interference effect-time (SIE-T), and SIE correct count (SIE-C). The TUGT was categorized into four stages: getting up (GT), walking straight (WT), turning around (TT), and sitting down (ST), in which the total time of TUGT was calculated. Results: After two weeks of treatment, the evaluation indexes were improved in the two groups, some of which were statistically significant. In the experimental group, SCWT-T, SIE-T, SIE-C, GT, WT, TT, ST, and TUGT were significantly improved after treatment (*p* < 0.05). SCWT-C, L-FMA score, 10MWT, GT, WT, stride length, step width, foot plantar pressure, pressure center curve, and activities of daily living were not statistically different from those before treatment (*p* > 0.05). After treatment, SCWT-T, SIE-C, SIE-T, BBS score, TT, and ST in the experimental group were significantly shorter than those before treatment, with statistical differences (*p* < 0.05). Compared with the sham group, SCWT-C, L-FMA score, 10MWT, GT, WT, TUGT, stride length, step width, foot plantar pressure, pressure center curve, and motor skills were not significantly improved (*p* > 0.05). Conclusion: It was revealed that post-stroke rTMS treatment of patients with ED could improve executive function, improve postural control function, and reduce the risk of falling. In addition, rTMS of DLPFC could be a therapeutic target for improving postural control ability and reducing the risk of falling.

## 1. Introduction

The need for post-stroke care due to motor dysfunction and decreased activities of daily living remarkably limits patients’ ability to participate in society and poses a great economic burden to their families and society. Decreased postural control after stroke is the main factor affecting patients’ daily mobility capability and quality of life. The incidence of post-stroke falling is still noticeable. It has been reported that up to 65% of patients have fallen during hospitalization, and 73% of patients have experienced falling at home or in the community within 6 months after stroke [1]. 

In our previous study, we found that patients with reduced executive function (EF) had poorer balance and postural control ability and were at a higher risk of falling [2]. EFs reflect a series of cognitive processes that are necessary for the control of cognitive behaviors, including decision-making, planning, cognitive flexibility, attention, working memory, etc. [3]. These important mental abilities can assist people to adapt to complex conditions. When EF is impaired, patients cannot make plans and perform self-adjustment according to rules, remarkably hindering patients’ attendance in their families and society. Repetitive transcranial magnetic stimulation (rTMS) is a noninvasive neuromodulation therapeutic technique that uses a series of magnetic stimuli to target brain areas to modulate the excitability of neurons in the cortex at the stimulated site. Which can produce long-term potentiation (LTP) or long-duration long-term depression (LTD) [4]. It is generally believed that low-frequency rTMS are stimuli with frequency < 1 Hz, while high-frequency rTMS have frequency > 5 Hz. Low-frequency rTMS reduces cortical excitability, while high-frequency rTMS upregulate dermal excitability [5]. After a stroke, rTMS can promote functional recovery by inhibiting the unimpaired excitability of the injured motor cortex or increasing the impaired hemispheric cortex. 

Studies have shown a consistent improvement in cognitive function with rTMS. It has not been reported that improving post-stroke EF can enhance walking and postural control abilities. However, whether improving executive function can improve motor function and postural control ability, and whether TMS treatment on the dorsolateral prefrontal cortex can improve motor function and postural control after improving executive function has not been reported. In this study, we tried to improve patients’ executive function by regulating DLPFC with rTMS and observed whether the patients’ motor function, gait, balance, and postural control ability were improved while the patients’ executive function was improved. Through this research, we ultimately want to provide a new treatment concept for improving patients’ walking and postural control ability. 

## 2. Methods

### 2.1. Participants

A total of 18 patients with stroke who were admitted to the Department of Rehabilitation Medicine, Beijing Tiantan Hospital Affiliated to Capital Medical University (Beijing, China) from September 2021 to March 2022 were enrolled (like Figure 1). The inclusion criteria were as follows: (1) patients aged 35–65 years old; (2) right-handed patients; (3) patients with a history of the first cerebral hemorrhage or cerebral infarction in basal ganglia; (4) ≤3 months post-stroke; (5) patients with educational level of junior high school or above; (6) 15 ≤ MOCA score ≤ 25; (7) ability to walk more than 10 meters independently. The exclusion criteria were as follows: (1) patients with aggravated/unstable cerebrovascular disease; (2) patients with a history of complications, such as cerebral hemorrhage due to trauma, and other neurological or psychological diseases; (3) with infarction in other brain area or white matter lesions; (4) basal ganglia hemorrhage but no softening foci in the external capsule; (5) patients with serious heart, lung, liver, or renal dysfunction or with malignant tumors; (6) patients with sensory aphasia, abnormal sensory and other cognitive domains, anxiety and depression before or after stroke, and those who were unable to cooperate with assessment and treatment; (7) a history of epilepsy, or a family history of epilepsy; (8) the deterioration of conditions, and the emergence of new infarction or a large area of cerebral infarction. The following criteria were considered for trial suspension: (1) participants with severe adverse reactions or being unable to continue their participation in the study; (2) deterioration of the conditions or serious complications; (3) participants who did not cooperate or did not receive treatment according to regulations; (4) leaving the study by participants and their family members.

A total of 18 patients were finally enrolled in this study. Using the random number table method, patients were assigned into 2 groups (*n* = 9 cases in each group). In the treatment group, there were 7 men and 2 women who were aged (54.6 ± 11.83) years old, including 5 cases of cerebral infarction and 4 cases of cerebral hemorrhage (3 cases of left-sided lesions and 6 cases of right-sided lesions). They had been educated for 9–15 years. In the sham group, there were 8 men and 1 woman who were aged (57.37 ± 12.78) years old, including 5 cases of cerebral infarction and 4 cases of cerebral hemorrhage (5 cases of right-sided lesions and 4 cases of left-sided lesions). There was no statistically significant difference in baseline data between the two groups (*p* < 0.05). 

All patients signed informed consent. The study was approved by the Ethics Committee of Beijing Tiantan Hospital Affiliated to Capital Medical University (Approval No. KY2021-040-02).

### 2.2. Study Design

A randomized, double-blind study was designed. Patients were randomly assigned into two groups, including rTMS group (experimental group) and sham group (control group). All patients were blinded to the treatment. Evaluators were also unaware of patients’ grouping. Patients in both groups received routine secondary prevention of cerebrovascular events and routine rehabilitation treatment. Standardized comprehensive rehabilitation treatment program is the use of nerve promotion technology, Bobath therapy, exercise relearning method, and traditional exercise therapy to comprehensively apply to the overall rehabilitation of stroke patients with hemiplegia. Each patient received rehabilitation training for 45 m/day. 

The rTMS protocol: in the experimental group, the “8” coil was positioned on the surface of the scalp of the left dorsolateral prefrontal cortex (DLPFC) projection area. The coil was placed vertically in the sham group [6]. Electromyographic electrode was used to record motor evoked potentials (MEPs) at the muscular abdomen of the first dorsal interosseous muscle. According to the guidelines of the International Federation of Clinical Neurophysiology published in 2012, when determining the threshold of resting exercise, subthreshold stimulation should be used to start with the initial detection of 35% maximum output intensity (MOI), and the stimulus intensity gradually increased. Generally, 5% MOI gradually increases, which can continuously lead to MEP with wave amplitude > 50 µV. Then, the stimulus intensity gradually decreases until no more than 5 effective MEPs in 10 stimuli, and an additional 1% increase in the output intensity indicates the resting motor threshold (MT). In the present study, stimulation frequency was 5 Hz, and it was attempted to use a stimulus intensity of 80% MT and 1200 pulses for 5 days/week for 2 weeks. If patients feel uncomfortable, the treatment was immediately stopped, and it was essential to indicate whether there were any uncomfortable symptoms 20 min after the treatment.

At present, the relevant parameters of TMS treatment for motor function, cognitive function, and sensory function after stroke are not unified, and the results are mostly effective, but there is no comparison of the efficacy of two different parameters. According to “Evidence-based guidelines on the therapeutic use of repetitive transcranial magnetic stimulation (rTMS): An update (2014–2018)” [7], Mainly studies choose the stimulation frequency was 5 Hz–20 Hz, stimulus intensity of 80–110% MT and 600–2000 pulses for 5 days/week for 2–4 weeks”. Considering that our enrolled patients were all patients in the early stage of stroke, we conservatively chose the prescription of the disclosed dose and frequency, also taking the advice from some experts.

### 2.3. Assessments 

Screening: the scores of MoCA and the Stroop color-word test (SCWT) were utilized as the outcome measures of EF. The lower limb Fugl–Meyer assessment of (L-FMA) was used to assess the lower limb motor function. The 10-m walking test (10MWT), gait analysis, and plantar pressure analysis were employed to assess the gait and postural control ability. Berg balance scale (BBS) and timed up and go test (TUGT) were used to assess the balance and postural control function. 

MoCA scoring was developed by Nasreddine et al. [8]. It is a simple screening tool to assess cognitive and attentional/executive functions, and it has also been used in studies on executive function assessment [9]. SCWT was used to measure executive function. It has three cards: card A included 50 color words (yellow, red, blue, and green); card B consisted of dots in four colors (yellow, red, blue, and green); card C covered four color words, which were printed in four different colors (yellow, red, blue, and green). In lieu of thinking about the meaning of each word, subjects were asked to identify color of each word as quickly and accurately as possible. Examiner recorded the time of each card (SCWT-T) and the correct number (SCWT-C). Stroop interference effect (SIE) was calculated as follows: SIE-time (SIE-T) = time of card C − time of card B; SIE correct count (SIE-C) was calculated as follows = correct number of card B − correct number of card C. The greater of the SIE, the worse the interference inhibition function, and the worse the EF.

Analysis of gait and plantar pressure: using the Zebris FDM 1.12 measuring system, subjects were asked to wear tight clothing, and thin socks on test board with their upper limbs swinging in a natural rhythm. “START” was clicked, and data were collected 30 s after adaptation. The following parameters of walking cycle were extracted: step speed, stride length, step width, foot plantar pressure, the peak values of forefoot and rear foot pressure, the length of support line in front and back directions of COP in a single support period, and the difference between the left and right symmetries. 

In the TUGT, as described previously [2], the TUGT was categorized into the four stages: getting up (GT), walking straight (WT), turning around (TT), and sitting down (ST), and the total time of TUGT was also recorded.

### 2.4. Statistical Analysis 

Statistical analysis was performed using GraphPad Prism 8.0 software (GraphPad Software, Inc., San Diego, CA, USA). The basic data were presented by frequency, constituent ratio, mean and standard deviation, etc., and the continuous variables were statistically described by mean ± standard error. The measurement data satisfied normal distribution and homogeneity of variance. The paired-sample *t*-test was used for intra-group comparison at different time points, the independent sample *t*-test was employed for inter-group comparison at the same time point, and the chi-square test was utilized for the analysis of count data. A two-sided *p* < 0.05 was considered statistically significant. 

## 3. Results 

### 3.1. Baseline Data 

There was no statistically significant difference between the two groups in the baseline data before treatment (*p* > 0.05) (Table 1). 

### 3.2. SCWT before and after Treatment 

Before treatment, there was no significant difference in SCWT score between the two groups (*p* > 0.05). After 2 weeks of treatment, SCWT-T, SIE-T, and SIE-C in the experimental group were significantly improved compared with those before treatment (*p* < 0.05). SCWT-C was elevated, while there was no statistical significance (*p* > 0.05). In the sham group, SCWT-T, SCWT-C, SIE-C, and SIE-T were not significantly improved compared with those before treatment (*p* > 0.05). In the experimental group, significant differences were found in SCWT-T, SIE-C, and SIE-T after treatment compared with those in the sham group (*p* < 0.05), while no significant difference was noted in SCWT-C compared with that in the sham group (*p* > 0.05) (Table 2).

### 3.3. L-FMA Score before and after Treatment 

Before treatment, there was no statistically significant difference in L-FMA score between the two groups (*p* > 0.05). After weeks of treatment, the L-FMA scores in the experimental group were not significantly elevated compared with those before treatment (*p* > 0.05). There was no significant difference in L-FMA score between the two groups before and after treatment (*p* > 0.05) (Table 3). 

### 3.4. MWT and BBS Scores before and after Treatment

Before treatment, there was no significant difference in the 10MWT and BBS scores between the two groups (*p* > 0.05). After two weeks of treatment, the BBS score in the experimental group was significantly higher than that before treatment (*p* < 0.05). There was no significant difference in the 10MWT score before and after treatment (*p* > 0.05). After treatment, the BBS score in the experimental group was significantly higher than that in the sham group (*p* < 0.05). There was no significant improvement in the 10MWT score between the two groups, and there was no statistical difference in the 10MWT and BBS scores in the sham group before and after treatment (*p* > 0.05, Table 4).

### 3.5. TUGT Score before and after Treatment 

Before treatment, there were no significant differences in GT, WT, TT, ST, and TUGT between the two groups (*p* > 0.05). After two weeks of treatment, the total time of GT, WT, and TUGT in the experimental group was not significantly different from that before treatment (*p* > 0.05). In addition, the duration of TT and ST in the experimental group was significantly shortened after treatment compared with that before treatment (*p* < 0.05). There were no significant differences in WT, TT, ST, TT, ST, and TUGT in the sham group compared with those after treatment (*p* > 0.05). After treatment, in the experimental group, the duration of TT and ST significantly decreased compared with that in the sham group (*p* < 0.05). Moreover, GT, WT, and TUGT showed no significant improvement compared with the sham group (*p* > 0.05) (Table 5).

### 3.6. Gait and Foot Plantar Pressure Parameters before and after Treatment 

Stride length, step width, and COP are the line from front to back and bilateral symmetry (the difference between left and right lateral COP trajectories) in a single support period. There was no significant difference between the two groups before and after treatment (*p* > 0.05). After treatment, there were no statistically significant differences in stride length, step width, and COP in the anteroposterior distance and the left and right symmetries between the two groups (*p* > 0.05) (Table 6).

## 4. Discussion

The results of the present study suggested that high-frequency rTMS stimulation of DLPFC could improve the response time and anti-interference ability of patients. Although no significant changes were observed in gait or L-FMA, balance and postural control were significantly improved, and the risk of falling during independent daily activities was significantly reduced.

At present, there is no perfect treatment plan for EF rehabilitation. No controlled study has concentrated on the efficiency of treatment strategies, and no report has guided clinicians in the selection of strategies for individual cases. It has been shown that patients who received rTMS (5~20 Hz at 80%~110% MT) of the left DLPFC could significantly improve ED patients’ EF after stroke. rTMS is a non-invasive brain stimulation technique that applies pulsed magnetic field to the brain to cause neuronal excitation or inhibition, thereby affecting brain metabolism and electrical activity. Studies of animal models reported that the beneficial effect of rTMS may be induced by the upregulation of neurotrophic or growth factors [10], and rTMS can improve CI in dementia model rats by changing the activity of N-methyl-d-aspartic acid receptor and brain-derived neurotrophic factor (BDNF) [11]. Studies have shown that cortical plasticity is enhanced after rTMS, which is related to inhibitory cortical circuits and BDNF upregulation in different functional brain regions [12]. The increased concentration of BDNF in the cerebral cortex may complete synaptogenesis and promote the formation and branching of dendritic spines, thus promoting cortical functional remodeling in stroke patients [13].

The cognitive dysfunction after stroke is characterized by impairment of memory, attention, executive ability, and social behavior. In an epidemiological cohort study on chronic brain injury, Jaillard et al. [14] reported that the most common cognitive symptoms were memory impairment (90%), attention impairment (82%), and ED (75%). Different incidence rates of ED have been reported, depending on the cognitive domain tested and the definition of executive function. Post-stroke ED may be ignored under perfect environmental conditions in hospitals with the care of nursing, doctors, therapists, and barrier-free facilities. When they return to their families, communities, or social occupational activities, their disabilities may be revealed. ED was not detected at the early stage and missed treatment time, resulting in more serious consequences [3].

EF plays a critical role in post-stroke recovery and has a high risk of functional dependence. Some patients are unable to perform different tasks and others cannot inhibit erroneous or unrelated behaviors. The majority of patients are only able to complete a single step of a complex problem, while they are unable to present the right solution. In addition, ED patients are mainly unable to return to work and have poor social participation ability [15].

EF in the early stage of a stroke may seriously affect the recovery of motor function [16]. A previous study showed that patients with cognitive dysfunction had a worse motor function recovery one year after stroke. The number of patients with ED who could not recover their motor function was four times higher than that without ED [17]. A cross-sectional study of 20 patients with stroke revealed that patients with ED performed worse on a complex walking test compared with patients with a normal EF [18]. The results of the present study showed that ED had an effect on walking ability with complex postural changes, such as turning and sitting, while it slightly influenced walking straight, which was evidenced by the results of 10MWT, TUGT, and gait analysis. 

Disruption of dorsolateral prefrontal subcortical circuitry leads to ED [19]. A previous study [20] showed that the MoCA score decreased after continuous theta-burst stimulation (cTBS) damage to the left DLPFC after temporary virtual injury. However, right DLPFC stimulation did not affect the task performance. The present study suggested that the left DLPFC is associated with EF. Liu-Ambrose et al. [21] confirmed that ED was independently associated with poor balance, mobility, and exercise endurance in patients with chronic stroke. A longitudinal study reported that participants with a poor EF at 3 months and 1 year after a stroke had significantly lower levels of balance and physical activity than participants without these impairments [20]. From a clinical point of view, individuals who exhibit deficits in the management of their thoughts and activities after stroke may have difficulty organizing a family exercise program at discharge [22]. Our previous study also confirmed that patients with a poor ED in the early stage stroke had poor postural control and balance, and balance function and postural control ability were significantly correlated with EF. Horak et al. [23] hypothesized that individuals with cognitive dysfunction may more frequently use existing cognitive processing methods to posture control and falling may be caused by insufficient cognitive processing to posture control, while they are busy with secondary cognitive tasks. The present study showed that patients with improved EF in the TUGT could significantly enhance the indicators of turning around and sitting in the secondary task during walking.

Wagner et al. [24] found that the Stroop task and Wisconsin card sorting test were significantly improved after dorsolateral prefrontal rTMS treatment. Another study reported that [25] the cognitive function index was significantly elevated in patients with mild cognitive impairment who received rTMS (10 Hz at 120% of MT) of the left DLPFC. The results of this study revealed that SCWT-T, SIE-C, and SIE-T of patients significantly improved after 2 weeks of rTMS treatment. In addition, with the improvement of EF, the patient’s postural control was significantly enhanced.

According to previous reports, executive dysfunction can hinder the recovery of motor function. People with ED have worse balance and a higher risk of falling. Therefore, in this study, we designed to observe the changes in motor function, gait, balance, and postural control in patients with executive dysfunction before and after treatment. However, the results of this study showed that there was no significant difference between the two groups before and after treatment in gait or L-FMA. This result may suggest that the mechanism of executive function on motor function and gait is different from that of postural control, or that there is little effect of executive function on motor function.

In the present study, it was found that the inhibitory control ability of patients who received high-frequency rTMS of the left DLPFC after stroke was significantly improved, their anti-interference ability was ameliorated, and their postural control ability was raised during walking. With the improvement of EF, patients’ abilities to turn around and perform sit-to-stand transfers were significantly improved, BBS was significantly ameliorated, and the risk of falling was significantly reduced. Although it was proven that improved cognition could reduce patients’ risk of falling, the gait, and COP were not significantly improved in the present study, which could be related to the sample size or treatment duration. Additionally, in order to avoid the effects of biomechanical factors (abnormal muscle strength, muscle tone, movement pattern) on poor control of posture, patients’ L-FMA scores were generally high, and the abnormal movement patterns were not obvious, thus, no significant difference was found in the FMALE score after treatment. Gait and COP may be more influenced by biomechanics, and rhythmic walking requires fewer cognitive resources [2]. Executive function has a greater impact on postural control than motor function and gait. After the left DLPFC rTMS, motor function and gait were not significantly improved with the improvement of executive function, but postural control ability and balance ability were significantly changed.

## 5. Conclusions

In summary, ED increases the risk of falling. Moreover, post-stroke rTMS treatment of the left DLPFC can improve EF, enhance postural control function, and reduce the risk of falling. It was revealed that rTMS of DLPFC may be a therapeutic target for improving postural control ability and attenuating the risk of falling.

## 6. Limitations and Prospects 

There are some limitations to the study. It is difficult to collect participants in order to ensure the consistency of the affected brain regions, further study is still needed because of the limited cases allotted. In the study, we used the modality to place the coil vertically in the sham group, which does not ensure adequate simulated stimulation—a sham coil would be better. That may also have affected the results. In this study, all patients were in the early stage of stroke, and we conservatively chose the prescription of the disclosed dose and frequency. Further studies are needed to determine whether there is a better treatment option. We will improve them in further research to obtain more meaningful results and guide clinical practice better. 

## Figures and Tables

**Figure 1 brainsci-12-01185-f001:**
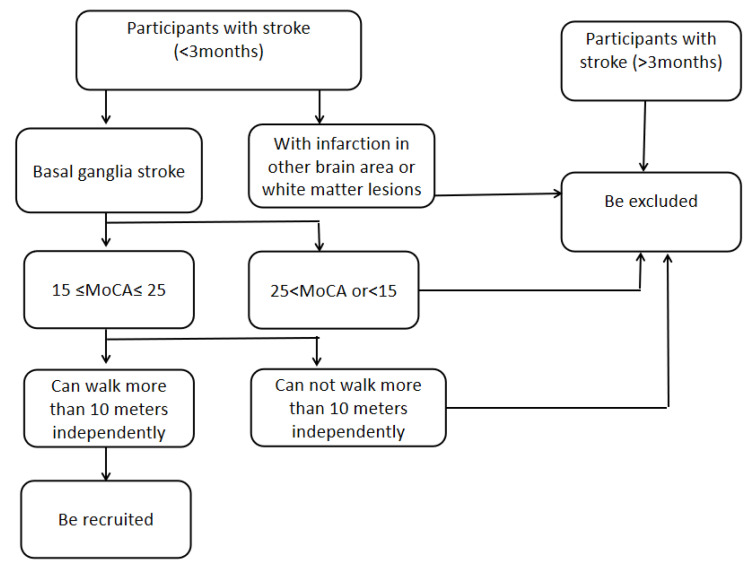
Recruitment flow chart.

**Table 1 brainsci-12-01185-t001:** The baseline data.

Group	Age (Year)	Sex (*n*)	Hemiplegic Limb (*n*)	Onset Time (Month)	Type (*n*)
Sham group	57.37 ± 12.78	F 1 M 8	L 4 R 5	1.34 ± 0.27	CH 4 CI 5
Experimental group	54.6 ± 11.83	F 2 M 7	L 3 R 6	1.01 ± 0.32	CH 4 CI 5

CH: cerebral hemorrhage; CI: cerebral infarction.

**Table 2 brainsci-12-01185-t002:** SCWT and SIE test.

Group	SCWT-T (s)				SCWT-C (*n*)			
	Before	After	T	*p*	Before	After	T	*p*
Sham group	130.36 ± 26.78	118.78 ± 38.34	2.35	0.36	134.90 ± 2.38	140.34 ± 43.25	−2.72	0.23
Experimental group	125.98 ± 31.70	99.26 ± 18.62	5.43	0.006 #	126.47 ± 3.69	146.73 ± 43.25	3.42	0.09
T	1.58	5.69			1.32	−2.79		
*p*	0.67	0.009 *			0.59	0.25		
**Group**	**SIE-T (s** **)**				**SIE-C (*n*)**			
	**Before**	**After**	**T**	** *p* **	**Before**	**After**	**T**	** *p* **
Sham group	35.36 ± 16.78	28.78 ± 11.26	1.98	0.48	11.05 ± 4.27	8.79 ± 2.52	2.47	0.37
Experimental group	41.03 ± 9.37	15.37 ± 8.04	5.79	0.002 †	10.74 ± 3.39	4.28 ± 0.98	5.78	0.002 †
T	2.36	5.21			1.47	6.79		
*p*	0.35	0.004 ‡			0.49	0.001 ‡		

Before: before treatment; after: after treatment; #: SCWT-T in the experimental group was significantly improved compared with those before treatment (*p* < 0.05); *: In the experimental group, SCWT-T was significant differences compared with those in the sham group after treatment (*p* < 0.05); †: SIE-T and SIE-C in the experimental group were significantly improved compared with those before treatment (*p* < 0.05); ‡: In the experimental group, SIE-T and SIE-C were significantly improved compared with those in the sham group after treatment (*p* < 0.05).

**Table 3 brainsci-12-01185-t003:** The lower limb Fugl–Meyer assessment.

Group	Before Treatment	After Treatment	T	*p*
Sham group	20.27 ± 7.37	29.82 ± 9.25	−1.98	0.48
Experimental group	24.01 ± 8.27	31.95 ± 10.28	−1.70	0.51
T	−1.99	−1.38		
*p*	0.43	0.51		

There was no significant difference in L-FMA score between the two groups before and after treatment (*p* > 0.05); there was no significant difference in L-FMA score in each group before and after treatment (*p* > 0.05).

**Table 4 brainsci-12-01185-t004:** The 10MWT and BBS score.

Group	10MWT (s)				BBS			
	Before	After	T	*p*	Before	After	T	*p*
Sham group	21.57 ± 10.36	16.47 ± 6.36	2.01	0.31	41.73 ± 12.32	46.79 ± 12.51	−2.56	0.39
Experimental group	19.28 ± 7.61	16.28 ± 4.23	1.98	0.42	39.69 ± 13.47	52.37 ± 13.48	−5.12	0.002 #
T	1.27	0.96			2.03	−4.79		
*p*	0.57	0.64			0.35	0.011 *		

There was no significant difference in 10MWT score between the two groups before and after treatment (*p* > 0.05); there was no significant difference in 10MWT score in each group before and after treatment (*p* > 0.05). #: The score of BBS in the experimental group was significantly improved compared with those before treatment (*p* < 0.05); *: in the experimental group, the score of BBS was significantly higher than those in the sham group after treatment (*p* < 0.05).

**Table 5 brainsci-12-01185-t005:** TUGT (s).

	Group	Before Treatment	After Treatment	T	*p*
GT	Sham group	3.27 ± 0.25	2.92 ± 0.56	1.35	0.5
	Experimental group	4.01 ± 0.82	2.95 ± 0.22	1.65	0.51
T		1.90	−0.8		
*p*		0.63	0.81		
WT	Sham group	12.84 ± 4.3	9.85 ± 2.31	1.78	0.42
	Experimental group	14.01 ± 5.24	10.05 ± 4.26	1.81	0.39
T		−1.61	−1.23		
*p*		0.48	0.77		
TT	Sham group	4.27 ± 1.34	3.81 ± 0.85	1.75	0.48
	Experimental group	5.01 ± 1.21	1.95 ± 0.68	4.85	0.01 #
T		−1.99	5.38		
*p*		0.43	0.00 *		
ST	Sham group	3.83 ± 0.31	2.03 ± 0.45	2.18	0.07
	Experimental group	3.90 ± 0.63	1.29 ± 0.08	4.97	0.01 #
T		−1.12	4.63		
*p*		0.63	0.01 *		
TUGT	Sham group	24.36 ± 7.36	21.36 ± 9.28	2.24	0.28
	Experimental group	26.21 ± 8.39	20.95 ± 8.2	4.29	0.02 #
T		−1.64	1.47		
*p*		0.33	0.45		

#: Time of the TT, ST, TUGT was significantly shorter compared with those before treatment in the experimental group (*p* < 0.05); *: in the experimental group, the time of ST, TT was significantly higher than those in the sham group after treatment. TUGT: time of stand up and go test; GT: time of getting up; WT: time of walking straight, TT: time of turning around, ST: time of sit down (*p* < 0.05).

**Table 6 brainsci-12-01185-t006:** Gait analysis and plantar pressure parameters.

	Group	Before Treatment	After Treatment	T	*p*
Stride (cm)	Sham group	35.21 ± 10.24	40.97 ± 16.4	1.37	0.53
	Experimental group	34.01 ± 12.34	42.95 ± 14.52	1.67	0.48
T		1.02	−0.87		
*p*		0.83	0.76		
Step width (cm)	Sham group	14.85 ± 6.34	11.85 ± 4.31	1.48	0.52
	Experimental group	16.01 ± 5.72	12.05 ± 6.26	1.61	0.39
T		−1.05	−1.03		
*p*		0.68	0.77		
Front and rear support lines (cm)	Sham group	14.27 ± 1.34	16.81 ± 0.85	−1.25	0.57
	Experimental group	15.01 ± 1.21	18.95 ± 0.68	−1.75	0.41
T		−1.38	−1.49		
*p*		0.43	0.58		
Bilateral symmetry (cm)	Sham group	4.83 ± 0.31	4.03 ± 0.45	1.68	0.17
	Experimental group	5.42 ± 0.63	3.79 ± 0.08	2.07	0.09
T		−1.18	1.63		
*p*		0.52	0.31		

There was no significant difference in gait analysis and plantar pressure parameters between the two groups before and after treatment (*p* > 0.05); there was no significant difference in gait analysis and plantar pressure parameters in each group before and after treatment (*p* > 0.05).

## Data Availability

The data presented in this study are openly available in “Clinical Trial Management Public Platform”(Number: ChiCTR2200055412). Also, the data can be available on request from the corresponding author.

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
