# Peer review of "Effects of Repetitive Transcranial Magnetic Stimulation on Gait and Postural Control Ability of Patients with Executive Dysfunction after Stroke"

_brainsci, 2022, doi:10.3390/brainsci12091185_

Round 1

Reviewer 1 Report

The manuscript describes the effect of rTMS on improving symptoms of executive function in patients with stroke. The idea of using rTMS is not new and the manuscript fails to address as to what is novel about this approach. There is also sections of the introduction and discussion that merits rewriting to define the study.

The introduction should include literature citations that have studied impact of high frequency vs low frequency rTMS effects and the rationale for each.

“According to the guidelines of the International Federation of Clinical 113 Neurophysiology published in 2012” Are there more current guidelines in rTMS treatment? Are the rTMS parameters different for motor vs sensory vs mood deficits?

The manuscript will benefit from citing :  Fan, Huiliu, et al. "The effect of repetitive transcranial magnetic stimulation on lower-limb motor ability in stroke patients: a systematic review." Frontiers in Human Neuroscience 15 (2021).

“Although no significant changes were observed in gait or L-FMA” This was the focal point of the entire study What was the rationale for this?

The discussion should involve dissecting the results. It is not clear from the discussion on what the authors have to state.

Author Response

Point 1: The manuscript describes the effect of rTMS on improving symptoms of executive function in patients with stroke. The idea of using rTMS is not new and the manuscript fails to address as to what is novel about this approach. There is also sections of the introduction and discussion that merits rewriting to define the study.

Response 1: Thanks very much for the professional advice. We apologize for the lack of clarity in our description. We have supplemented the innovation points of this study in the introduction and discussion section according to your suggestions.

Executive dysfunction after stroke can affect the recovery of motor function and postural control. No relevant studies have been reported about improving postural control by improving executive function. Repeated transcranial magnetic stimulation of the left dorsolateral prefrontal lobe can improve executive function in patients with this condition, which has been verified by many studies. Previous studies have shown that TMS stimulation of motor brain areas can improve motor function. However, whether TMS treatment on dorsolateral prefrontal cortex can improve motor function and postural control after improving executive function has not been reported. In this study, the method of using transcranial magnetic stimulation to improve patient executive function was not innovative. The innovation of this study is to try to improve the motor function and postural control of patients by improving executive function through transcranial magnetic stimulation(Line 69).

Point 2: The introduction should include literature citations that have studied impact of high frequency vs low frequency rTMS effects and the rationale for each.

Response 2: Thanks very much for the important advice.

 We have supplemented the literature citations in the introduction and discussion section according to your suggestions.

Point 3: “According to the guidelines of the International Federation of Clinical 113 Neurophysiology published in 2012” Are there more current guidelines in rTMS treatment? Are the rTMS parameters different for motor vs sensory vs mood deficits?

Response 3: Thanks very much for professional question.

 At present, the relevant parameters of TMS treatment for motor function, cognitive function and sensory function after stroke are not unified, and the results are mostly effective, but there is no comparison of the efficacy of two different parameters. The following table provides evidence for the treatment parameters of TMS using cognitive function as an example according to “Evidence-based guidelines on the therapeutic use of repetitive transcranial magnetic stimulation (rTMS): An update (2014–2018)”.

Target spot

Intensity

of stimulation

Frequency

Pulse

Intensity of treatment

course

Type

DLPFC

110%RT

5HZ

2000

5次/w

2w

II

Herwig et al., 2001; Fitzgerald et al., 2009; Ahdab et al., 2010 

120%RT

10HZ

3000

5次/w

2w

III

Padala et al., 2018

80%RT

5HZ

1200

5次/w

4w

II

Wu et al. (2015)

80-100%

20HZ

2000

5次/w

4w

III

Rutherford et al., 2015

Point 4: The manuscript will benefit from citing: Fan, Huiliu, et al. "The effect of repetitive transcranial magnetic stimulation on lower-limb motor ability in stroke patients: a systematic review." Frontiers in Human Neuroscience 15 (2021).

Response 4: Thanks very much for professional comments.

This review reporte that using rTMS can stimulate the representative brain areas of lower limb movement and further help patients improve motor function and gait. There are many methods we can lear.  In our study, rTMS can stimulate the left DLPFC can improve the EF, posture control, but the motor function and gait. Maybe in our future research, we can use rTMS stimulate combined DLPFC and motor brain regions to Improve the patient's motor ability, gait and postural control ability, so as to better improve the patient's ability of daily living.

Point 5: “Although no significant changes were observed in gait or L-FMA” This was the focal point of the entire study What was the rationale for this?

Response 5: Thanks very much for professional comments.

According to previous reports, executive dysfunction can hinder the recovery of motor function. People with ED have worse balance and a higher risk of falling. Therefore, in this study, we designed to observe the changes of motor function, gait, balance and postural control in patients with executive dysfunction before and after treatment. However, the results of this study showed that there was no significant difference between the two groups before and after treatment in gait or L-FMA. This result may suggest that the mechanism of executive function on motor function and gait is different from that of postural control. Or there little effect of executive function on motor function(We also supplement this in the discussion section of the manuscript( line 342).

Point 6: The discussion should involve dissecting the results. It is not clear from the discussion on what the authors have to state.

Response 6: Thanks very much for the important advice.

We have refined the discussion section according to your suggestions.

Reviewer 2 Report

This manuscript was to investigate the effects of repetitive transcranial magnetic stimulation (rTMS) on gait and postural control ability of patients with executive dysfunction (ED) after stroke.This filed is interesting., and the data is abundant. However, the revised version will have a better impact in the area and the readers of the journal will have a better understanding of this manuscript.

1.The manuscript still needs some minor copyediting for grammatical errors, typos, and word omissions.  

2. The necessary references should be marked. For instance, In the section of Introduction, After stroke, rTMS can promote functional recovery by inhibiting the unimpaired excitability of the injured motor cortex or increasing the impaired hemispheric cortex.

Studies have shown a consistent improvement in cognitive function with rTMS. It has not been reported that improving poststroke EF can enhance walking and postural control abilities. “

3.Whether the consent inform is signed by the patients is not mentioned in the present manuscript.

4.In the section of Discussion, the animal experiments on rTMS need to be mentioned to discuss the possible molecular mechanisms for the clinical effect of rTMS.

5.In the whole, some paragraphs have poor coherence among different sentences in the section of Discussion, so more Cohesive words need to be used approximately.

6.The future outlook and possible research direction should be summarized at the end of the manuscript.

Author Response

This manuscript was to investigate the effects of repetitive transcranial magnetic stimulation (rTMS) on gait and postural control ability of patients with executive dysfunction (ED) after stroke.This filed is interesting., and the data is abundant. However, the revised version will have a better impact in the area and the readers of the journal will have a better understanding of this manuscript.

Point 1: The manuscript still needs some minor copyediting for grammatical errors, typos, and word omissions.

Response 1: Thanks very much for the professional advice.We are very sorry for our mistakes. And we have corrected the grammatical errors, typos, and word omissions carefully.

Point 2: The necessary references should be marked. For instance, In the section of Introduction, “After stroke, rTMS can promote functional recovery by inhibiting the unimpaired excitability of the injured motor cortex or increasing the impaired hemispheric cortex.

Response 2: Thanks very much for the important advice.

 We have supplemented the literature citations in the introduction and discussion section according to your suggestions. 

Point 3: Studies have shown a consistent improvement in cognitive function with rTMS. It has not been reported that improving poststroke EF can enhance walking and postural control abilities.

Response 3:  Thanks very much for the professional advice. We apologize for the lack of clarity in discussion.

 Executive dysfunction after stroke can affect the recovery of motor function and postural control. No relevant studies have been reported about improving postural control by improving executive function. Repeated transcranial magnetic stimulation of the left dorsolateral prefrontal lobe can improve executive function in patients with this condition, which has been verified by many studies. Previous studies have shown that TMS stimulation of motor brain areas can improve motor function. However, whether TMS treatment on dorsolateral prefrontal cortex can improve motor function and postural control after improving executive function has not been reported. In this study, the method of using transcranial magnetic stimulation to improve patient executive function was not innovative. The innovation of this study is to try to improve the motor function and postural control of patients by improving executive function through transcranial magnetic stimulation

Point 4: Whether the consent inform is signed by the patients is not mentioned in the present manuscript.

Response 4:  Thanks very much for the professional advice.

We have supplemented this content according to your suggestions(line 114).

Point 5: In the section of Discussion, the animal experiments on rTMS need to be mentioned to discuss the possible molecular mechanisms for the clinical effect of rTMS.

Response 5: Thanks very much for professional advice.

We have supplemented this content in discussion section(line 281).

Point 6: In the whole, some paragraphs have poor coherence among different sentences in the section of Discussion, so more Cohesive words need to be used approximately.

Response 6: Thanks very much for professional comments.

We have reviewed and revised the full text

Point 7: The future outlook and possible research direction should be summarized at the end of the manuscript.

Response 7: Thanks very much for the important advice.

We have upplemented this content at the end of the manuscript.

Reviewer 3 Report

The authors deal with a relevant and timely topic, i.e., the effects of 2-week repetitive transcranial magnetic stimulation (rTMS) of the left dorsolateral prefrontal cortex (DLPFC) on gait and postural control of 18 patients with executive dysfunction after stroke, compared to conventional rehabilitation and sham (fictitious) stimulation. Globally, they concluded that post-stroke rTMS treatment of patients with executive dysfunction could improve them and reduce the risk of falling. Overall, the paper is nicely conceived; the results seem to be consistent and are adequately discussed. Some comments, especially in the Methods.

Abstract: please provide more details on both rTMS (e.g., site, frequency) and sham. The same holds true for randomization, patients’ blindness, and operators’ blindess (with respect to the outcome measures).

Introduction: among clinical applications of rTMS within the proposed topic, please briefly mention the use of rTMS in cognitive impairment (for a recent comprehensive review, please see PMID: 34482205).

Introduction: please clearly state the experimental hypothesis: what did the authors expect? Why? How?

Methods: inclusion criteria appear to be quite broad, e.g., age from 35 to 65 years (different recovery capacity between young and older patients) and both ischemic and hemorrhagic strokes (distinctive conditions, with very different outcome). The same holds true for exclusion criteria: “aggravated/unstable cerebrovascular disease” (not clear definition) and “history of complications, such as cerebral hemorrhage” (which, however, was mentioned among inclusion criteria). Please carefully check and revise accordingly.

Methods: the samples obtained after the randomization procedure are rather heterogeneous in terms of sex distribution (M/F), type of stroke (ischemic/hemorrhagic), and hemispheric side of lesion (right/left).

Methods: was the left DLPFC stimulated in all participants, regardless of right or left stroke? This may be inappropriate and should have preliminarly considered all the stroke features (type, side, site, extent, etc).

Methods: today, the modality to place the coil vertically in the sham group does not ensure an adequate simulated stimulation; a sham coil would have been needed. Please list it among the study limitations.

Methods: please refer to the latest guidelines of the International Federation of Clinical Neurophysiology, published in 2015 (PMID: 25797650). Moreover, the authors should justify the rTMS protocol adopted: “In the present study, stimulation frequency was 5 Hz, and it was attempted to use a stimulus intensity of 80% MT and 1200 pulses for 5 days/week for 2 weeks”. Please refer/compare to previously published protocols.

Results: please edit/revise this section, including tables and text, in order to make it easier to understand.

Discussion: among the possible mechanisms underlying the positive effects of rTMS observed in this study, the induction and modulation of metaplasticity should be mentioned and discussed (e.g., PMID: 34276553).

Discussion: please list the limitations of the study, current pitfalls, and research agenda.

Conclusions: due to some study limitations, the conclusions should be a bit softened (also in the Abstract).

General: although the language is acceptable, an editing by a native-English speaker would be helpful.

Author Response

The authors deal with a relevant and timely topic, i.e., the effects of 2-week repetitive transcranial magnetic stimulation (rTMS) of the left dorsolateral prefrontal cortex (DLPFC) on gait and postural control of 18 patients with executive dysfunction after stroke, compared to conventional rehabilitation and sham (fictitious) stimulation. Globally, they concluded that post-stroke rTMS treatment of patients with executive dysfunction could improve them and reduce the risk of falling. Overall, the paper is nicely conceived; the results seem to be consistent and are adequately discussed. Some comments, especially in the Methods.

Point 1: Abstract: please provide more details on both rTMS (e.g., site, frequency) and sham. The same holds true for randomization, patients’ blindness, and operators’ blindess (with respect to the outcome measures.

Response 1: Thanks very much for the professional advice.

 We have supplemented the contents according to your suggestions in abstract(line 18).  

Point 2: Introduction: among clinical applications of rTMS within the proposed topic, please briefly mention the use of rTMS in cognitive impairment (for a recent comprehensive review, please see PMID: 34482205).

Response 2: Thanks very much for the important advice.

We have supplemented the contents in the introduction section according to your suggestions(line ). 

At present, there are few studies on TMS targeting cognitive dysfunction after stroke. Most studies refer to mild cognitive impairment and dementia treatment guidelines such as the article PMID: 34482205.

The following table provides evidence for the treatment parameters of TMS using cognitive function as an example according to “Evidence-based guidelines on the therapeutic use of repetitive transcranial magnetic stimulation (rTMS): An update (2014–2018)”.

Target spot

Intensity

of stimulation

Frequency

Pulse

Intensity of treatment

course

Type

DLPFC

110%RT

5HZ

2000

5次/w

2w

II

Herwig et al., 2001; Fitzgerald et al., 2009; Ahdab et al., 2010 

120%RT

10HZ

3000

5次/w

2w

III

Padala et al., 2018

80%RT

5HZ

1200

5次/w

4w

II

Wu et al. (2015)

80-100%

20HZ

2000

5次/w

4w

III

Rutherford et al., 2015

Point 3: Introduction: please clearly state the experimental hypothesis: what did the authors expect? Why? How?

Response 3:  Thanks very much for the professional advice.

We have supplemented the contentsat the end of the introduction section according to your suggestions

Point 4: Methods: inclusion criteria appear to be quite broad, e.g., age from 35 to 65 years (different recovery capacity between young and older patients) and both ischemic and hemorrhagic strokes (distinctive conditions, with very different outcome). The same holds true for exclusion criteria: “aggravated/unstable cerebrovascular disease” (not clear definition) and “history of complications, such as cerebral hemorrhage” (which, however, was mentioned among inclusion criteria). Please carefully check and revise accordingly

Response 4:  Thanks very much for the professional advice.

We apologize for the mistakes we made in the text, and we have We made some supplements in the exclusion criteria section. We recruited patients taking into account the age and the different pathologic mechanisms and outcomes of ischemia and hemorrhage.We performed a comprehensive analysis of the imaging and symptoms of the enrolled patients. Patients with infarction in other brain area, with basal ganglia hemorrhage but no softening foci in the external capsule, without executive dysfunction, patients with other cognitive domain disorders, and age-affected white matter lesions were all ruled out to ensure the reliability of the research results.

Point 5: Methods: the samples obtained after the randomization procedure are rather heterogeneous in terms of sex distribution (M/F), type of stroke (ischemic/hemorrhagic), and hemispheric side of lesion (right/left).

Response 5: Thanks very much for professional advice.

The incidence of stroke is very different between men and women, with men significantly more than women. It has also been reflected in many studies.  we also consulted some experts abou this heterogeneity, they think it can be acceptable if make sure there is no difference between the two groups at baseline. There were no difference between the two groups in the type of stroke at baseline.

Point 6: Methods: was the left DLPFC stimulated in all participants, regardless of right or left stroke? This may be inappropriate and should have preliminarly considered all the stroke features (type, side, site, extent, etc).

Response 6: Thanks very much for professional comments.

In this study, all patients were received rTMS at left DLPFC referring to other studies. A large number of studies have shown that the left DLPFC is associated with executive function, while the right DLPFC is not. Therefore, TMS therapy aimed at improving executive function after stroke is mostly selected as the therapeutic target.

Point 7: Methods: today, the modality to place the coil vertically in the sham group does not ensure an adequate simulated stimulation; a sham coil would have been needed. Please list it among the study limitations.

Response 7: Thanks very much for the important advice.

We have supplemented this content at the end of the manuscript.

Point 7: Methods: please refer to the latest guidelines of the International Federation of Clinical Neurophysiology, published in 2015 (PMID: 25797650). Moreover, the authors should justify the rTMS protocol adopted: “In the present study, stimulation frequency was 5 Hz, and it was attempted to use a stimulus intensity of 80% MT and 1200 pulses for 5 days/week for 2 weeks”. Please refer/compare to previously published protocols. 

Response 3: Thanks very much for professional question. 

In the latest guidelines of the International Federation of Clinical Neurophysiology, published in 2015 (PMID: 25797650), We studied the detailed mechanism and procedure of TMS treatment. It provides a high reference value for clinical practice.

At present, the relevant parameters of TMS treatment for motor function, cognitive function and sensory function after stroke are not unified, and the results are mostly effective, but there is no comparison of the efficacy of two different parameters. According to “Evidence-based guidelines on the therapeutic use of repetitive transcranial magnetic stimulation (rTMS): An update (2014–2018)”(PMID:31901449), Mainly studies choose the stimulation frequency was 5Hz-20Hz, stimulus intensity of 80%-110% MT and 600-2000 pulses for 5 days/week for 2-4 weeks”. Considering that our enrolled patients were all patients in the early stage of stroke, we conservatively chose the prescription of the disclosed dose and frequency. We also take advice from some experts. With further research, we hope we can find better treatment prescriptions.

Point 8: Results: please edit/revise this section, including tables and text, in order to make it easier to understand.

Response 8: Thanks very much for the important advice.

We have supplemented this content at the end of the manuscript.

Point 8: Discussion: among the possible mechanisms underlying the positive effects of rTMS observed in this study, the induction and modulation of metaplasticity should be mentioned and discussed (e.g., PMID: 34276553).

Response 9: Thanks very much for the important advice.

We have supplemented this content in the discussion.

Point 10: Discussion: please list the limitations of the study, current pitfalls, and research agenda.

Conclusions: due to some study limitations, the conclusions should be a bit softened (also in the Abstract).

Response10: Thanks very much for the important advice.

We have supplemented this content at the end of this article.

Point 11: General: although the language is acceptable, an editing by a native-English speaker would be helpful.

Response11: Thanks very much for the important advice.

This article has been proofread and edited for proper English language, grammar, punctuation, spelling, and overall style by one or more of the qualified scientific editors at MedSci. We will provide a certification of English editing or indicate that a revised version has been edited by a native English speaker when resubmitting the revised version.

Reviewer 4 Report

Dear authors:

His work has great merit and is very well done, however there are some aspects that need to be improved.

in line 78 the 4 does not go as in the rest of the manuscript

the introduction should include more content and better justification of the study, could you add more content?

The results tables are too close together and the characters are not well appreciated, therefore they should be improved.

It would be much more graphic to introduce a small flowchart so that the reader finds it easier to follow the process of selecting patients and the reasons for their exclusion.

Author Response

Please see the attachment below

Round 2

Reviewer 1 Report

The author(s) have addressed my queries. 

Reviewer 3 Report

In this revised version, the authors have adequately addressed most of the previously raised concerns, thus significantly improving the quality of this manuscript.  However, they did not considered some of the literature suggested, which, instead, would have enriched both the background and the discussion sections of this manuscript.